Epiperipatus puri sp. nov., a new velvet worm from Atlantic Forest in Southeastern Brazil (Onychophora, Peripatidae)

Costa Cristiano Sampaio 1
http://orcid.org/0000-0002-7220-6396 Mendes Amanda Cruz 2
Giupponi Alessandro Ponce de Leão 3 agiupponi@gmail.com
1 Departament of Biology and Zoology, Universidade Federal de Mato Grosso—UFMT , Cuiabá, Mato Grosso , Brazil
2 Departament of Zoology, Universidade do Estado do Rio de Janeiro , Rio de Janeiro , Brazil
3 Fundação Oswaldo Cruz—FIOCRUZ, Instituto Oswaldo Cruz—IOC, Collection CAVAISC, LAC , Rio de Janeiro , Brazil
Gillespie Joseph
Electronic publication date: 2023 Oct 2
Publication date: 2023
Volume: 11
Electronic Location ID: e15384
Received 2022 Mar 10; Accepted 2023 Apr 18
Copyright: © 2023 Costa et al.
Copyright year: 2023
Copyright holder: Costa et al.
License: This is an open access article distributed under the terms of the Creative Commons Attribution License, which permits unrestricted use, distribution, reproduction and adaptation in any medium and for any purpose provided that it is properly attributed. For attribution, the original author(s), title, publication source (PeerJ) and either DOI or URL of the article must be cited.
License URL: https://creativecommons.org/licenses/by/4.0/

Keywords: New species, Neotropical, Neopatida, Taxonomy, Biodiversity, Threatened species

Funding: São Paulo Research Foundation FAPESP #2011/20211-0, 2012/02969-6 and 2014/20557-2 This work and fieldwork for Cristiano Sampaio Costa was funded by the São Paulo Research Foundation (FAPESP, who was supported by fellowships from FAPESP #2011/20211-0, 2012/02969-6 and 2014/20557-2). The funders had no role in study design, data collection and analysis, decision to publish, or preparation of the manuscript.

==============================
Epiperipatus ohausi (Bouvier, 1900) is the first species known from Rio de Janeiro, and more than 120 years later a new species is described in the state of Rio de Janeiro (RJ). In this study, we describe the second species in the state of Rio de Janeiro, which we are naming in honor of the indigenous population called puri who resided in southeastern coastal Brazil. The species can be diagnosed mainly by large dorsal primary papillae close to the insertion of the legs drawing a light band from the anterior to the posterior region of the body, and large dorsal primary papillae alternating on the dorsal plicae. Moreover, they are recognized in vivo by the color of the diamond-shaped marks brownish orange on the dorsal portion of the body. Epiperipatus puri sp. nov. morphologically seems to be related to Epiperipatus acacioi (Marcus & Marcus, 1995) by the shape of the primary papillae apical piece and to E. ohausi by the resemblance of dorsal papillae. The phylogeny shows a close relationship between the new species and E. ohausi in a clade with a still undescribed species from Rio de Janeiro, Brazil located within the Atlantic Forest, one of the most threatened biomes in the world.

Introduction

Currently, the circa 200 recent species of Onychophora are distributed in two families, Peripatidae Evans, 1901 and Peripatopsidae Bouvier, 1905. The phylum Onychophora (velvet worms) has received more attention in the last decade than ever, especially the clade Neopatida (=the Neotropical Peripatidae), which resulted in the recent description of five species (Baker et al., 2021).

Although velvet worms are among the most fascinating terrestrial groups of invertebrates, their biodiversity is poorly understood, and the taxonomy is elusive (Sedgwick, 1888; Read, 1988; Giribet et al., 2018). Some causes are the poor sampling of individuals in the field, low numbers of specimens available in museums often accompanied by inadequate preservation, poor access to historical collections, scarce type-locality data, and the conservative morphology of the group (Froehlich, 1968; Peck, 1975; Read, 1988; Chagas-Jr & Costa, 2014; le Bras et al., 2015; Costa & Giribet, 2016; Costa, Chagas-Jr & Pinto-da-Rocha, 2018; Giribet et al., 2018). The lack of distinguishable external characters capable of clearly delimiting the boundaries of the taxa has led to the search for new methodologies such as scanning electron microscopy (e.g., Read, 1988; Morera-Brenes & Monge-Nájera, 1990; de Oliveira, Wieloch & Mayer, 2010; Costa & Giribet, 2016) and DNA sequences analyses which have led to the description of many species for the phylum in the last decade (e.g., de Oliveira et al., 2013; Giribet et al., 2018)⁠.

Epiperipatus Clark, 1913 is the most diverse of the 11 recent genera within Peripatidae, with 36 valid species distributed in the Antilles, Central, and South America, with 15 species from Brazil (Peck, 1975; Brito et al., 2010; de Oliveira et al., 2011; de Oliveira, Read & Mayer, 2012; Costa, Chagas-Jr & Pinto-da-Rocha, 2018; Costa, Giribet & Pinto-da-Rocha, 2021).

Epiperipatus was erected by Clark (1913) as a subgenus of Peripatus Guilding, 1826 along with other two new subgenera (Plicatoperipatus Clark, 1913 and Macroperipatus Clark, 1913), designating Peripatus edwardsii Blanchard, 1847 as the type species and transferring more nine species to this subgenus (Clark, 1913:⁠ 18). Since Peck (1975)⁠, Epiperipatus is treated as genus, although this author did not explicitly elevate the rank from the subgenera of Peripatus. Peck cited it under an identification key for families and genera of Onychophora and made the combinations with the species without using Peripatus in the binomina (Peck, 1975⁠: 345). After Peck’s work Epiperipatus counted with 17 species. Subsequent works described new species as Epiperipatus (Brito et al., 2010; de Oliveira et al., 2011; Costa, Chagas-Jr & Pinto-da-Rocha, 2018) or reassigned extant ones to the genus (de Oliveira, Wieloch & Mayer, 2010; Chagas-Jr & Costa, 2014; Costa, Chagas-Jr & Pinto-da-Rocha, 2018; Costa, Giribet & Pinto-da-Rocha, 2021; Morera-Brenes & Monge-Nájera, 2010).

A recent revision of Epiperipatus intended to improve the delimitation of the genus based on morphological and molecular data, aiming to test the information of 33 morphological characters applied in the taxonomy of Peripatidae (Costa, Giribet & Pinto-da-Rocha, 2021)⁠. In this article, although Epiperipatus appears as non-monophyletic, a core monophyletic group emerges including the type species Epiperipatus edwardsii, which was recently redescribed under modern parameters (Costa, Chagas-Jr & Pinto-da-Rocha, 2018)⁠.

An undescribed species occurs in Cachoeiras de Macacu, Rio de Janeiro, Brazil, and according to Costa, Giribet & Pinto-da-Rocha (2021) belongs to Epiperipatus. The area where the types were collected is a fragment of the Atlantic Forest (AF, hereafter), an extremely threatened biome (Ribeiro et al., 2011). The AF is one of the 36 conservation hotspots of the world (Myers et al., 2000; CEPF, 2019), and one of the most important given its conservation value in the area (Laurance, 2009)⁠.

In the present work, this species is newly described as Epiperipatus puri sp. nov. Its type specimens are deposited at Museu Nacional, Universidade Federal do Rio de Janeiro, (MNRJ), as part of the rebuilding of the burned MNRJ zoological collections. Epiperipatus puri sp. nov. is the first species of onychophorans deposited in MNRJ after its main building was lost in a ruthless fire. The collection of Onychophora was held together with the collections of Arachnida and Myriapoda, under the curatorship of Adriano B. Kury. Most of the three collections were lost in the fire except for material under loan and all the data, which were safe due to the routine backup policy of the curator (Kury, Giupponi & Mendes, 2018).

Materials and Methods

The type-series is composed of six specimens deposited in Museu Nacional, Universidade Federal do Rio de Janeiro, Rio de Janeiro (MNRJ) (curator A. B. Kury) and Museu de Zoologia da Universidade de São Paulo, São Paulo (MZUSP) (curator R. Pinto-da-Rocha), both in Brazil. The specimens were collected between 2012 and 2018 in a small patch of tropical humid forest (IBGE, 2019) (Fig. 1) around the Metropolitan Region of the state of Rio de Janeiro, under grass roots (Fig. 2F), in a private conservation area (Reserva Ecológica de Guapiaçu). License code number: INEA (Brasil) n° 005/2020 (Alessandro Ponce de Leão Giupponi). Specimens were preserved in 70% and 100% EtOH. We examined their morphology in detail and compared it with specimens of Epiperipatus ohausi (Bouvier, 1900) from Nova Iguaçu municipality, Rio de Janeiro, Brazil to detect diagnostic features of the new species: MNRJ 0056; 1♀; BRAZIL, Rio de Janeiro, Nova Iguaçu, Reserva Particular do Patrimônio Natural dos Petroleiros; 23.XII.2009; Costa, C.S., Giupponi, A.P.L. leg. MNRJ 0058; 1♂; same locality; 11.III.2010; Costa, C.S., Chagas-Jr, A., Giupponi, A.P.L., Kury, A.B. leg.

Figure 1 Updated distribution of onychophorans in Brazil assembling the results of Costa, Chagas-Jr & Baptista (2009), Costa, Chagas-Jr & Pinto-da-Rocha (2018), and Costa, Giribet & Pinto-da-Rocha (2021).

The records of specimens of onychophorans in Guapiaçu represented by red stars in all images of this plate, state of Rio de Janeiro. We do not included Epiperipatus tucupi (Froehlich, 1968) in the maps due to the imprecise location information provided on the label, which mentions only “Pará.” The Reserva Ecológica de Guapiaçu (type-locality of Epiperipatus puri sp. nov.) is represented by a yellow star (image from Google Earth).

Figure 2 Photos of female paratype.

Epiperipatus puri sp. nov., body background and papillae of female paratype MZUSP 0122, and the environment where the specimens were found. Body background in (A and B) dorsal side, (C and D) lateral side and (E) ventral side. (F) Site where the specimens were collected, under the roots of the grass, at Reserva Ecológica de Guapiaçu (REGUA), Cachoeiras de Macacu, Rio de Janeiro, Brazil. Scales bars 2, 4 and 5 = 1 mm; 3, 6 = 2 mm.

We studied one of the specimens (MNRJ 0093, voucher 065) using scanning electron microscopy (SEM) following Chagas-Jr & Costa (2014) and Costa, Chagas-Jr & Pinto-da-Rocha (2018). We dissected out one mandible, the fifth oncopod of the left side, and a small rectangular section of the dorsal integument from the dorsomedian furrow to the base of the oncopods. The structures were critical point dried and mounted in SEM stubs with bi-adhesive carbon tape. A 5-nm gold layer was applied. Samples were imaged with a JEOL JSM-6390LV at the SEM Platform Rudolf Barth at Instituto Oswaldo Cruz—Fundação Oswaldo Cruz (IOC-FIOCRUZ).

Photographs in vivo were taken with a SONY Cybershot DSC-HX1 with built-in flash, or Canon EOS Rebel XS with a macro lens and flash circular camera. Images were edited using Adobe Photoshop CS5 (Figs. 2A–2E). For color descriptions, we followed the standard names of the 267 Color Centroids of the NBS/ISCC Color System (see Kelly, 1958; also available at Centore, 2016, and W3Schools, 2023). Also, we combined the stereomicroscopy and Scanning Electron Microscopy (SEM) studies of the external morphology of the specimens for descriptions. The morphological descriptive nomenclature follows the terminologies of Read (1988)⁠, Morera-Brenes & Monge-Nájera (2010)⁠, de Oliveira, Wieloch & Mayer (2010)⁠, and Costa, Chagas-Jr & Pinto-da-Rocha (2018)⁠. All measurements are given in millimeters (mm). The species we described here was part of a more detailed study of Epiperipatus, combining morphological and molecular data for specimens from the Neotropics (Costa, Giribet & Pinto-da-Rocha, 2021).

The electronic version of this article in Portable Document Format (PDF) will represent a published work according to the International Commission on Zoological Nomenclature (ICZN). This published work and the nomenclatural acts it contains have been registered in ZooBank, the online registration system for the ICZN. The ZooBank LSIDs (Life Science Identifiers) can be resolved, and the associated information viewed through any standard web browser by appending the LSID to the prefix http://zoobank.org/. The LSID for this publication is urn:lsid:zoobank.org:pub:A8971D25-1C9D-4807-BE2B-730FB010B717. The online version of this work is archived and available from the following digital repositories: PeerJ, PubMed Central and CLOCKSS.

Results

Taxonomic results

Family Peripatidae Evans, 1901

Genus Epiperipatus Clark, 1915

Epiperipatus puri sp. nov.

Epiperipatus [sp6]: Costa: 2016; Costa, Giribet and Pinto-da-Rocha: 2021: 6, 8, 25.

(Figs. 1–4)

Figure 3 SEM of holotype.

Epiperipatus puri sp. nov., female holotype MNRJ 0093, scanning electron micrographs. (A) Dorsal plicae. The dorsomedian furrow position indicated by the arrow head, Additionally, two primary papillae are marked in purple, and three accessory papillae are highlighted in yellow. (B) shape of primary papilla, in lateral view, with the posterior view depicted on the left side of the image. (C) Foot of the fifth oncopodod in prolateral view, (D and E) on the top in inner and outer jaws blades respectively. Scales bars 3A = 500 μm; 3B = 50 μm; 3C–E = 100 μm.

Figure 4 Total evidence analysis adapted from Costa, Giribet & Pinto-da-Rocha (2021) of Epiperipatus puri sp. nov. and related species. Our species, represented by three vouchers, is in a clade in shades of blue with a so far unnamed species (Epiperipatus sp5: MNRJ 0059) and E. ohausi (MNRJ 0056, 0058), all from Rio de Janeiro (RJ), Brazil. The RJ clade is sister-group to a clade in orange composed by E. machadoi (MNRJ 0043) from Minas Gerais, Brazil and three unnamed species from Espírito Santo, Brazil (Epiperipatus sp13: MNRJ 0018, 0020, 0023, 0026; Epiperipatus sp.: MNRJ 0016, 0100; Epiperipatus sp11: MNRJ 0042, 0046).

Diagnosis (based on a combination of characters as follows). Epiperipatus species with large dorsal primary papillae close to the insertion of the legs drawing a narrow light band from the anterior to posterior regions of the body. The distribution of large pale dorsal primary papillae alternates the dorsal plicae with clear differences in the number in two sequential plicae. Moreover, they are recognized in vivo by the color the diamond-shape marks brownish-orange (Figs. 2A–2D).

Description of female holotype (MNRJ 0093).

Measurements. Length 43; width 3.0, height 3.9.

Color (for living specimen Figs. 2A–2E). Background color of body dark reddish-brown, overlaid by blurry diamond-shaped marks brownish orange. A broad dashed-line (close to the insertion of the legs) similarly colored as the diamond-shape marks. Dorsomedian furrow dark reddish-brown. Anterior portion of head moderate reddish-brown and antennae strong brown. Color of the dorsal portion of the legs grayish reddish orange. Legs and ventral surface display the same color, moderate reddish-brown. Ventral and preventral organs moderate brown.

Description of the body (Figs. 2A–2E and 3). Conspicuous dorsomedial furrow and hyaline organs along the main body axis (Figs. 2B and 3A). Twelve plicae per segment, two incompletes as broad as the diamond-shape marks, and seven crossing to the ventral side (Fig. 3A).

Almost all dorsal papillae on the plicae, except for the smaller accessory papillae on the furrow between the plicae. Primary dorsal papillae aligned on top of folds; two primary papillae separated by one to five accessory papillae, the former do not occur close together (Figs. 2D and 3A). Dorsal primary papillae with a conical basal piece composed of scales that never overlap each other at the base of the papillae (Figs. 3A and 3B). Primary papillae as the largest dorsal papillae, with roundish dome insertion and asymmetrical regular spherical apical piece (Fig. 3B). Basal piece larger than apical, with a range of at least seven scale ranks (Fig. 3B). Apical piece with three posterior scale ranks (Fig. 3B).

Narrow scales both on base and apical piece (Fig. 3B). Needle-shaped sensory bristle directed posteriorly (Fig. 3B). Small and large primary papillae with a conspicuous constriction between the base and the apical piece (Figs. 3A and 3B). Two-sized dorsal primary papillae: the largest are on the top of the body in continuous plicae (close to the dorsomedian furrow and drawing the diamond-shaped marks) and close to the legs. Lateral papillae in alternated dorsal plicae. Accessory papillae are the smallest dorsal, with roundish insertion similar in shape to the base of the primary papillae; they are more abundant per plicae and differ in position in relation to primary papillae (Figs. 2D and 3A).

Head. No evident structures or patterns on the head. Antennae (Figs. 2A–2C) composed of 40 rings. Antennal tip composed of seven broad rings, excluding the disc on the top, followed by a sequence of narrow and broad rings alternating until at least ring 20. Eyes and frontal organs present ventrolateral to the antennal base. Conspicuous frontal organs as long as six fused antennal papillae. Mouth opening surrounded by a small, anterior, unique lobe, and seven flanked lobes decreasing in size from the anterior to the posterior end of the mouth. Dental formula of inner and outer jaws (Figs. 3D and 3E), respectively: 1/1 and 1/2/10. The accessory tooth is thinner in the outer jaw. The second accessory tooth is reduced.

Legs. 28 pairs of legs in the holotype (Fig. 2E). Nephridial tubercle on fourth and fifth pairs of legs, between third and fourth spinous pads (Fig. 2E), connected by the top to the third spinous pad. On the fourth and the fifth pairs of legs, four spinous complete pads are present and no evidence of a fifth one.

Sexual dimorphism. Two pairs of pregenital legs with one crural papilla (male) each, absent in females. Anal glands are inconspicuous and represented only by two pores on the anterior margin of the anal aperture, absent in females.

Type material. Holotype: MNRJ 0093, 1♀, BRAZIL, Rio de Janeiro, Cachoeiras de Macacu, Reserva Ecológica de Guapiaçu (REGUA), X.2012, A.P.L. Giupponi, J.S. Silva leg. Paratypes: MNRJ 0087, 1 unsexed specimen, MNRJ 0088, 1♂, same locality, 28.II – 02.III.2012, A.P.L. Giupponi, J.S. Silva leg; MNRJ 0107, 1 unsexed specimen, same locality, 19.III.2018, R.L.C. Baptista leg; MZUSP 0122, 1♀, same locality, 21.XII.2014, A. Ferreira, A.P.L. Giupponi, A. Rezende, C.S. Costa leg.

Distribution. Only known from the type locality (Fig. 1).

Etymology. The epithet puri (in apposition) refers to the Puri indigenous group belonging to the Macro-Jê linguistic group. They inhabited, among other places, the mountain region of the Rio de Janeiro state where specimens of this species were collected. Noun in apposition.

Remarks. Paratype. Length 12 to 22; width 1.0 to 2.5. Legs. 26 (female) and 27 (male) pairs of legs.

Discussion

Classification of Epiperipatus puri sp. nov.

The fuzzy generic limits due to the lack of clear morphological characters is a major issue of Neopatida taxonomy. Splitting species of Peripatidae in several genera might be a problem, especially in the systematics of Brazilian fauna (Giribet et al., 2018: 860; Costa, 2016)⁠. Epiperipatus is one of the most speciose genera of Peripatidae (de Oliveira, Read & Mayer, 2012; Costa, Chagas-Jr & Pinto-da-Rocha, 2018) with 36 described species after this article.

However recent studies based on the molecular data regards the genus as non-monophyletic (Costa, 2016; Giribet et al., 2018)⁠. Phylogenetically, Costa, Giribet & Pinto-da-Rocha (2021) based on the study of four molecular markers besides morphological data distinguished two clades of Brazilian species, the smallest with species from the state of Pará and the largest with the remaining species of the country included in the study. Epiperipatus puri sp. nov. appears nested in the largest clade closely related with species from the state of Rio de Janeiro: Epiperipatus ohausi (Bouvier, 1900) and a potentially undescribed species (see in Costa, Giribet & Pinto-da-Rocha (2021): 23, fig. 3, clade S, and Fig. 4 here). This clade is sister-group to another composed by Epiperipatus machadoi (Oliveira & Wieloch, 2005) and three other unnamed species from the state of Espírito Santo, Brazil, which is geographically close to the state of Rio the Janeiro and located within the Atlantic Forest.

Epiperipatus puri sp. nov. is characterized as a new species by the roundish insertion of dorsal papillae, the three posterior scale ranks, and two prolateral and one retrolateral foot papillae in the feet of the fourth and fifth oncopods (Fig. 3C). The presence of incomplete folds differs E. puri sp. nov. from Epiperipatus brasiliensis (Bouvier, 1899), Epiperipatus tucupi Froehlich, 1968 and Epiperipatus cratensis Brito et al., 2010. The new species differs from Epiperipatus diadenoproctus de Oliveira et al., 2011 by the inconspicuous anal glands in E. puri sp. nov.

In Epiperipatus paurognostus de Oliveira et al., 2011 the background color of the body is reddish-brown (in vivo) and the fourth spinous pad can be complete or incomplete, in E. puri sp. nov. the background color of the body is dark reddish-brown (in vivo) and the fourth spinous pad is complete. The apical piece is conical in Epiperipatus adenocryptus (de Oliveira et al., 2011) and E. machadoi, conical/cylindrical in Epiper ipatus lucerna Costa, Chagas-Jr & Pinto-da-Rocha, 2018 and Epiperipatus marajoara (Costa, Chagas-Jr & Pinto-da-Rocha, 2018), and spherical in E. puri sp. nov. The apical piece is conical and reduced in Epiperipatus beckeri (Costa, Chagas-Jr & Pinto-da-Rocha, 2018) and Epiperipatus titanicus (Costa, Chagas-Jr & Pinto-da-Rocha, 2018), robust in Epiperipatus hyperbolicus (Costa, Chagas-Jr & Pinto-da-Rocha, 2018), while the apical piece is regular in E. puri sp. nov. The new species seems to be closely related to Epiperipatus acacioi (Marcus & Marcus, 1955) by the shape of the apical piece of the primary papillae, however, E. puri sp. nov. primary papillae are lighter than other papillae. E. ohausi and E. puri sp. nov. bear dorsal papillae with similar shape and size, but the latter also bear accessory papillae on the flanks and uniform background color of body and oncopods.

Additionally, the results of Costa, Giribet & Pinto-da-Rocha (2021) include the species in a clade containing the type species of the genus, although the clade also includes species of other genera. Giribet et al. (2018) suggested that Caribbean species of Epiperipatus are closer to Peripatus than to the remaining “Epiperipatus”, however, this could be confirmed only by the inclusion of Peripatus juliformis Guilding, 1826 (type species of Peripatus) in Peripatidae analyses, neither of both studies included this species due to the lack of specimens in the collections adequate for the analyses.

Although the boundaries between Epiperipatus and Peripatus remain unclear, we cautiously preferred keeping the new species as Epiperipatus, as their putative closer species E. ohausi is currently classified. According to Bouvier (1905), the new species could not be a Peripatus due to the presence of only two pairs of pregenital oncopods with crural papillae, in Peripatus should be more than three. Also, according to Read (1988) it could not be classified as Peripatus because it has three ranks of scales in the apical piece, Peripatus should have more ranks.

Conservation

Epiperipatus puri sp. nov. is the 16th Epiperipatus species described from Brazil (Chagas-Jr & de Oliveira, 2019)⁠. Besides E. puri sp. nov., only E. ohausi is nominally recorded from Rio de Janeiro (See Costa, Chagas-Jr & Baptista, 2009; Costa, Chagas-Jr & Pinto-da-Rocha, 2018)⁠, a Brazilian state entirely within the Atlantic Forest domain.

The Atlantic Forest is the second largest rainforest in South America and one of the most distinctive biogeographic units in the Neotropical Region with high levels of endemism and biodiversity (Ribeiro et al., 2011). However, this biome has experienced large habitat losses since European colonization and currently, only 12.59% of its original area remains (Ribeiro et al., 2009, 2011) making it one of the “hottest hotspots” for conservation (Mittermeier et al., 2004; Laurance, 2009).

Our species was collected in a private conservation area, the RPPN Reserva Ecológica de Guapiaçu (REGUA) (Fig. 1). The reserve is in the upper Guapiaçu River Valley and started as a group of farms registered as a non-governmental organization in the early 2000s. Today part of its area (in a total of 357 ha of two areas) is officially recognized as one of the private conservation areas of the state of Rio de Janeiro, part of the state program of Private Reserves of Natural Patrimony (Portuguese acronym: RPPN) (see Guagliardi, 2018)⁠. The total area of the reserve (official and nonofficial) encompasses 7,500 ha of forest in different stages of forest cover, with an altitudinal range from 0 to above 2,000 m.a.s.l. (Soares et al., 2011). The region is in a mountain range where the Guapi-Macacu Basin belongs, which contributes to the water supply of 2.5 million inhabitants of five municipalities (Rodríguez Osuna et al., 2014). In this watershed, the degradation of aquatic resources is caused by urbanization, intense agriculture, and conversion of riparian vegetation (Rodríguez Osuna et al., 2014)⁠. The forest cover of the landscape in the Guapi-Macacu Basin is circa 40%, and it is a mosaic of different ages; unfortunately fragments anterior to 1976 occupy only 12% of the landscape (Costa et al., 2017)⁠.

Although composed mainly of secondary forests, REGUA is important to the conservation of local fauna. The area is known for its rich fauna of birds (Pimentel & Olmos, 2011)⁠, butterflies (Soares et al., 2011), mosquitoes (Silva et al., 2014), and for dragonflies and damselflies, with more than 200 species (Kompier, 2015; Pinto & Kompier, 2018)⁠. E. puri sp. nov. is the first Onychophora to REGUA. This demonstrates the high value of this reserve for the recovery of the endangered area it is embedded.

For more than 30 years, scientists advocated for the importance of invertebrates and their conservation, but the perspective has not changed much (Wilson, 1987; Collen et al., 2012). The velvet worms are at risk given their distribution in threatened biomes, such as the Atlantic Forest itself, and because they seem to occur in small-sized populations, although the amount of data available on population dynamics is scarce (New, 1995). Sometimes the newly described species are already critically endangered (e.g., de Oliveira et al., 2015)⁠.

Currently 80% of Brazilian species of onychophorans are in the Livro Vermelho da Fauna Brasileira Ameaçada de Extinção (ICMBio, 2018)⁠. One of the species considered endangered in Brazil’s Red Book is E. ohausi (see Costa, Cordeiro & Chagas-Jr, 2018), the only named species from Rio de Janeiro state before this work, considered here to be a close species to E. puri sp. nov.

Epiperipatus ohausi is known from Petrópolis (type-locality) and Nova Iguaçu (Chagas-Jr & Costa, 2014), forests from both localities suffer from pressures of urbanization. The population of the species is severely fragmented since it occupies humid shaded habitats, with an extent amount of litter, typical of forested areas (Costa, Cordeiro & Chagas-Jr, 2018)⁠. Although E. puri sp. nov. is distributed in a close area, also with high pressures of urbanization and agriculture, and probably its distribution extends to the area of Parque Estadual dos Três Picos, a State Reserve contiguous to REGUA. This reinforces the need for the preservation of those reserves and encourages the expansion of their areas.

After a few years of political stability and economic growth (1995-2014), with policies prioritizing the fight against poverty, environmental destruction, and historical deficit in science and education, Brazil is passing through severe economic, political, and social turmoil (Dobrovolski et al., 2018). Recently the indexes of deforestation are skyrocketing in all Brazilian biomes (Instituto Nacional de Pesquisas Espaciais (INPE), 2023), which seems to be related to recent policies favoring livestock ranching and agribusiness, and the weakening of the Brazilian system of protection of the environment and Indigenous lands (Ferrante, Gomes & Fearnside, 2020).

Although fires in Amazonian Forest, Cerrado, and Pantanal are usually related to the replacement of natural vegetation by cattle ranching and soy crops, in the AF the deforestation is related to urbanization (see Joly, Metzger & Tabarelli, 2014) and pressure of the real estate market. The increment in deforestation occurs in association with the negligence towards the scientific institutions (the destruction of Museu Nacional and its collections being an emblematic symbol), which suffered significant budget cuts (Martelli-Jr et al., 2019; Escobar, 2019), staff shortage, and direct federal political intervention in their management, which undermines the protection of fragile biota.

Conclusions

Our description of Epiperipatus puri sp. nov. contributes to the knowledge of the biodiversity in a hotspot for conservation, the Atlantic Forest. We characterized the species morphologically with the use of SEMs and photographs, including in vivo (important for recognizing the species in the field). All the type material was collected in a private reserve that is contiguous to a State Protected Area, demonstrating the importance of this type of initiative. E. puri sp. nov. was assigned to Epiperipatus such as the putative closer species, E. ohausi, but future studies could reveal the actual boundaries of the genera for there is molecular evidence that they could belong to Peripatus (Giribet et al., 2018)⁠. One of the known obstacles to the conservation of invertebrates is the poor state of knowledge of the species, many still unnamed. In the case of velvet worms, the difficulty to describe a species is notorious, and one recent proposed solution to deal with this problem is to connect information about undescribed species to common names (Sosa-Bartuano, Monge-Nájera & Morera-Brenes, 2018)⁠.

Supplemental Information

Supplemental Information 1 Epiperipatus puri table sequence data.

Click here for additional data file.

We are grateful to Plataforma de Microscopia Eletrônica Rudolf Barth (FIOCRUZ–IOC) and to their staff, Wendell Girard Dias, Roger Magno Macedo Silva, Rômulo Custódio dos Santos, and Taíssa Ribeiro Adriano de Oliveira, for the help with the SEM images. We are grateful to Adriano B. Kury (MNRJ) who provided support with the Latin grammatical forms. We are in debt to Júlia dos Santos Silva (FIOCRUZ) Nicholas Locke, and Raquel Locke who assisted with SEM images. Jorge Bizarro for valuable information about REGUA. We also thank Flávio Uemori Yamamoto for providing beautiful photographs of the new species. We also thank Alicia Santos and Jackie Thai (PeerJ Editorial Board) and two anonymous reviewers for their valuable contributions that enhanced the article.

Additional Information and Declarations

Competing Interests

Author Contributions

Field Study Permissions

Data Availability

New Species Registration

The authors declare that they have no competing interests.

Cristiano Sampaio Costa conceived and designed the experiments, performed the experiments, analyzed the data, prepared figures and/or tables, authored or reviewed drafts of the article, and approved the final draft.

Amanda Cruz Mendes analyzed the data, prepared figures and/or tables, authored or reviewed drafts of the article, and approved the final draft.

Alessandro Ponce de Leão Giupponi analyzed the data, prepared figures and/or tables, authored or reviewed drafts of the article, and approved the final draft.

The following information was supplied relating to field study approvals (i.e., approving body and any reference numbers):

INEA (Brazil) #005/2020 to Alessandro Ponce de Leão Giupponi.

The following information was supplied regarding data availability:

The sequences are available at NCBI:

- Epiperipatus [sp5] MNRJ 0059 ony-012: MN905635-MN544110.

- Epiperipatus [sp11] MNRJ 0046 ony-003: MN905628-MN544104.

- Epiperipatus [sp11] MNRJ 0042 ony-027a: MN905643, MN639368, MN544117.

- Epiperipatus [sp11] MNRJ 0042 ony-027b: MN905644, MN639369, MN544118.

- Epiperipatus [sp11], MNRJ 0042 ony-027c: MN905645, MN639370, MN544119.

- Epiperipatus [sp13] MNRJ 0020 ony-029: MN639371, MN544120.

- Epiperipatus [sp13] MNRJ 0018 ony-030a: MN639372, MN544121.

- Epiperipatus [sp13] MNRJ 0018 ony-030b: MN639373.

- Epiperipatus [sp13] MNRJ 0023 ony-048: MN905657, MN639383, MN544133.

- Epiperipatus [sp13] MNRJ 0026 ony-049: HG531958, MN933779, MN933788.

- Epiperipatus [sp14] MNRJ 0016 ony-032: MN544122.

- Epiperipatus machadoi MNRJ 0043 ony-008: MN905632, MN639360, MN544108.

- Epiperipatus ohausi MNRJ 0058 ony-010: MN905633.

- Epiperipatus ohausi MNRJ 0056 ony-011: MN905634, MN639361, MN544109, MN705441.

- Epiperipatus sp. MNRJ 0100 ony-153: MN639449, MN544194, MN705490.

The following information was supplied regarding the registration of a newly described species:

Publication LSID: urn:lsid:zoobank.org:pub:A8971D25-1C9D-4807-BE2B-730FB010B717.

Epiperipatus puri species LSID: urn:lsid:zoobank.org:act:5F941DB2-F54D-4E32-9337-0A90B26D99E7.

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
