# Peer review of "Epiperipatus puri sp. nov., a new velvet worm from Atlantic Forest in Southeastern Brazil (Onychophora, Peripatidae)"

_PeerJ, doi:10.7717/peerj.15384_

## Round 0.1 · original submission · Major Revisions

Dear Dr. Costa and colleagues:

Thanks for submitting your manuscript to PeerJ. I have now received three independent reviews of your work, and as you will see, the reviewers raised some minor concerns about the research. Despite this, these reviewers are optimistic about your work and the potential impact it will have on research studying onychophorans. Thus, I encourage you to revise your manuscript, accordingly, taking into account all of the concerns raised by the three reviewers.

While the concerns of the reviewers are relatively minor, this is a major revision to ensure that the original reviewers have a chance to evaluate your responses to their concerns. There are many suggestions, which I am sure will greatly improve your manuscript once addressed.

Please work on clarity throughout your manuscript. Please make sure all morphological data are easy to access.

The majority of these criticisms is highly constructive and should immensely improve your manuscript. Please also note that reviewers 1 and 3 have included marked-up versions of your manuscript.

Therefore, I am recommending that you revise your manuscript, accordingly, taking into account all of the issues raised by the reviewers.

I look forward to seeing your revision, and thanks again for submitting your work to PeerJ.

Good luck with your revision,

Best,

-joe

·

Basic reporting

I encountered some minor difficulties reading the text, I´ve included suggestions on how to improve some sentences, also I commented which ideas or paragraphs needed to be rearranged. Introduction and background show the appropriate context for the study. One figure needs to be improved; I attached my comments there. I wasn´t able to access the repositories used to store the morphological data (I don’t know if I missed the link or they´re included as supplementary material).

Experimental design

I consider it appropriate.

Validity of the findings

I consider the findings as valid and I´m glad that a new species of onychophora has been described.

Additional comments

I enjoyed reading the manuscript, all my comments are annotated in the PDF, my main suggestion is making the manuscript less reiterative with respect to some ideas, especially in the conservation section, some ideas need to be polished a little bit, but overall, a good study.

·

Basic reporting

Review:
My general statement is that the manuscript can be published after some corrections and improvements, particularly, reduction of one section, expansion of another and thorough review of the English grammar.
1. BASIC REPORTING
The basic question is: is the information presented enough to conclude that this is a valid new species?
I find it sufficient, the microscopic analysis and the comparison with possibly related species indicate that the species is new and valid.

Experimental design

2. EXPERIMENTAL DESIGN
The sample size of six specimens is acceptable because this is a rare group. Additionally, the taxonomic characters used are updated. The methods are standard for the group and are presented in enough detail.

Validity of the findings

3. VALIDITY OF THE FINDINGS
The description and conclusions are fine, but I tried the link to the backup data and got an error message (CFML Runtime Error).

Additional comments

4. General comments
On the one hand, I feel that the text sometimes digresses, for example, the comments about the fire that affected the National Museum in Rio de Janeiro, and the long section about conservation in Brazil, seem to be out of place in a species description. On the other hand, in today’s digital world, that extra text does not cost much more to publish and might, with time, become the only place where readers in the future can learn about those topics that have a long life in the scientific literature, but soon disappear from the news.
In the discussion, I would recommend expanding the analysis of the implications that these findings have for the taxonomy and systematics of velvet worms.

·

Basic reporting

The presented paper is a valuable contribution to onychophoran biodiversity. The descriptions, images, and supplementary material are great and the MS surely is suitable to be published in Peerj.
However, the text must be revised grammatically and semantically concerning the English. There are several mistakes and sentences that seem to be translated directedly from Portuguese without the proper concern for the semantical differences between both languages.

Experimental design

no comment

Validity of the findings

no comment

Additional comments

no additional comments

---

## Round 0.2 · accepted · Accept

Dear Dr. Costa and colleagues:

Thanks for revising your manuscript based on the concerns raised by the reviewers. I now believe that your manuscript is suitable for publication. Congratulations! I look forward to seeing this work in print, and I anticipate it being an important resource for groups studying onychophorans. Thanks again for choosing PeerJ to publish such important work.

Best,

-joe